# Assessing Urinary Para-Hydroxyphenylacetic Acid as a Biomarker Candidate in Neuroendocrine Neoplasms

**DOI:** 10.3390/ijms252212317

**Published:** 2024-11-16

**Authors:** Renato de Falco, Susan Costantini, Luigi Russo, Denise Giannascoli, Anita Minopoli, Ottavia Clemente, Salvatore Tafuto, Carlo Vitagliano, Elena Di Gennaro, Alfredo Budillon, Ernesta Cavalcanti

**Affiliations:** 1Laboratory Medicine Unit, Istituto Nazionale Tumori-IRCCS-Fondazione “G. Pascale”, 80131 Naples, Italy; renato.defalco@istitutotumori.na.it (R.d.F.); l.russo@istitutotumori.na.it (L.R.); denise.giannascoli@istitutotumori.na.it (D.G.); a.minopoli@istitutotumori.na.it (A.M.); 2Experimental Pharmacology Unit, Laboratories of Naples and Mercogliano (AV), Istituto Nazionale Tumori-IRCCS-Fondazione “G. Pascale”, 80131 Naples, Italy; s.costantini@istitutotumori.na.it (S.C.); carlo.vitagliano@istitutotumori.na.it (C.V.); e.digennaro@istitutotumori.na.it (E.D.G.); a.budillon@istitutotumori.na.it (A.B.); 3Sarcoma and Rare Tumors Unit, Istituto Nazionale Tumori-IRCCS-Fondazione “G. Pascale”, 80131 Naples, Italy; ottavia.clemente@istitutotumori.na.it (O.C.); s.tafuto@istitutotumori.na.it (S.T.)

**Keywords:** cancer, neuroendocrine, biomarker, tyrosine, chromogranin

## Abstract

The management of neuroendocrine neoplasms (NENs) involves the measurement of serum chromogranin A (s-CGA), serum neuro-specific enolase (s-NSE), and urinary 5-hydroxindolacetic acid (5-HIAA). Urinary para-hydroxyphenylacetic acid (u-pHPAA), a metabolite of tyrosine, has been proposed as a potential biomarker for these diseases. This study aims to evaluate the effectiveness of u-pHPAA and tyrosine as biomarkers. We measured the levels of s-CgA, s-NSE, u-5-HIAA, u-pHPAA, and tyrosine in blood or 24 h urine samples collected at baseline (T_0_) and after 1 year of follow-up (T_1_) from a limited cohort of patients enrolled at Istituto Nazionale Tumori-IRCCS-Fondazione “G. Pascale”. Biomarker values were normalized using the ratios between T_1_ and T_0_ values (T_1_/T_0_ parameters). The T_1_/T_0_ ratios for s-CgA and u-pHPAA were significantly associated with the outcome of death (*p* = 0.044 and *p* = 0.022, respectively). An ROC curve analysis demonstrated outstanding performances for these biomarkers (AUC = 0.958 and AUC = 1.00, respectively) and the Kaplan–Meier survival analysis showed significant Log-rank test results (*p* = 0.001 and *p* < 0.001, respectively). Additionally, T_0_ serum tyrosine correlated with the outcome of death (*p* = 0.044), with the ROC curve showing good performance (AUC = 0.958) and the Kaplan–Meier analysis yielding significant Log-rank test results (*p* = 0.007). Our study confirms the role of s-CgA in the management of NEN patients and highlights the potential roles of u-pHPAA and serum tyrosine as biomarkers. Further research is needed to validate our findings in larger populations.

## 1. Introduction

Neuroendocrine neoplasms (NENs) comprise a heterogeneous group of neoplasms arising from the neuroendocrine system that most commonly occur in the gastropancreatic and bronchopulmonary tract, although they can originate from different organs [1,2].

Usually, NENs exhibit a slow growth and good prognosis. However, some of these grow more rapidly, resulting in less survival rates; therefore, the management of these tumors represents a clinical challenge [3].

The World Health Organization’s classification system categorizes NENs into different grades and stages, reflecting their biological behavior and prognosis [4].

In detail, the most common types are the well-differentiated neuroendocrine tumors (NETs), which account for 80–90% of cases; the poorly differentiated neuroendocrine carcinomas (NECs), representing 10–20%; and the mixed neuroendocrine/non-neuroendocrine forms, also known as MiNEN. Sub-classification of NETs and NECs can be aided by evaluating mitotic count and the Ki67 index.

The diffuse neuroendocrine cells can secrete a wide range of amines and polypeptide hormones, and therefore they show a wide array of biological behavior and clinical manifestations. NENs have retained the ability to secrete amines, peptides, hormones, and molecules that can enter into the systemic circulation and be used as clinical biomarkers that, in conjunction with imaging, can be informative and guide clinical decision-making [5].

Despite extensive efforts to identify useful biomarkers for NENs, there remains an urgent need to develop markers that combine diagnostic accuracy with guiding therapeutic interventions and detecting early relapses [6].

Serum-specific biomarkers to certain types of neuroendocrine tumors (e.g., insulin, gastrin) are helpful indicators of tumor activity. In contrast, the most commonly used general biomarker, chromogranin A (CgA), is less reliable because elevated levels can occur in conditions unrelated to the presence of NENs thus limiting its clinical utility [7].

A specific biomarker for NENs is serotonin or its main metabolite, 5-hydroxyindoleacetic acid (5-HIAA), measured in serum and/or urine samples, which can be used as a prognostic marker of survival above all in patients with carcinoid syndrome [8,9,10].

Moreover, elevated levels of neuro-specific enolase (NSE) are mostly found in small-cell lung cancer and poorly differentiated NENs [11].

Currently, despite their limitations, such as poor specificity and sensitivity, which fail to fully capture the complexity of NENs, there is a consensus that a multi-analytical panel comprising serum CgA, serum NSE, and urinary 5-HIAA is required for NENs diagnosing and monitoring [7,12].

In this regard, we recently observed that urinary para-hydroxyphenylacetic acid (u-pHPAA), a metabolite of tyrosine physiologically present in the urine samples of healthy subjects, showed significantly increased levels related to a worsening clinical condition in NET patients, representing a promising prognostic biomarker in such diseases [13].

This study aims to verify the utility of u-pHPAA as a new biomarker that can be added to those commonly used in the management of NENs.

In addition, different studies have assessed the metabolomics profile of NET patients in order to identify clinically useful novel biomarkers and novel enriched metabolic pathways in NETs [14,15]. We performed untargeted serum metabolomic by ^1^H-NMR to identify the possible role of different metabolites such as tyrosine, purine, and glutamate.

## 2. Results

### 2.1. Population Characteristics and Death Outcome

The urine and serum samples of a group of fourteen patients affected by NEN Grading G1 and G2 enrolled within the IMMUNeOCT trial (EudraCT number 2017-001613-83) were collected at baseline and after 1 year of follow-up and classified into two groups based on their outcome (Table 1, Appendix A). Baseline patient characteristics were well balanced between the two groups, although there was a statistically significant difference in the gender proportion. A majority of the patients were female (n = 11) with a median age of 46 years (19–65), whereas male patients (n = 3) had a median age of 65 years (55–78). Metastasis was present in eight patients (57%). Nine patients were under a therapy regimen with somatostatin analogues (Octreotid, Lanreotid, Sandostatin), while five did not receive treatment. Disease progression was observed in four patients during follow-up, and two male patients died. The male gender was associated with death (*p* = 0.033), while neither BMI nor age were associated with death or the presence of metastasis. Disease progression was not related to BMI, age, sex, or gender.

### 2.2. Differences in Biochemical Markers Levels

We analyzed the serum levels of CgA and NSE as well as of 5-HIAA and pHPAA in the 24 h urine samples at baseline (T_0_) and after one year of follow-up (T_1_) (Table 2). We found that at T_0_, the s-CgA values were associated with the presence of metastasis (*p* = 0.038) and death (*p* = 0.022), while at T_1_, both the s-CgA and u-5-HIAA levels were associated with death (*p* = 0.022 and *p* = 0.044, respectively). Disease progression was not related to any biochemical markers. No significant associations were found for pHPAA at the T_0_ or T_1_ time points for death, metastasis, or disease progression.

Due to the possible outlier values at baseline, we normalized the biomarker results by calculating the ratios between the values obtained after 1 year of follow-up and at baseline (T_1_/T_0_) (Table 2).

The analysis performed on the normalized values reveals that s-CgA is still associated with death (*p* = 0.044), while the significance for u-5-HIAA levels is loss. Interestingly, the values of the T_1_/T_0_ ratio for the u-pHPAA results were associated with the outcome of death (*p* = 0.022). In order to discriminate which parameter (s-CgA or u-pHPAA) had a better performance, an ROC curve analysis for the outcome of death was carried out (Figure 1).

We found that T_1_/T_0_ s-CgA exhibited good performance (AUC = 0.958; Sensitivity: 100%; Specificity: 83%) with a cut-off of 1.49 and that T_1_/T_0_ u-pHPAA showed an excellent performance, with an AUC of 1.000 and both sensitivity and specificity of 100% with a cut-off of 5.57. T_1_/T_0_ s-NSE and T_1_/T_0_ u-5-HIAA ROC curves showed poor performance (AUC = 0.833 and 0.625, respectively), in line with the absence of association with the outcome of death (Appendix A). Kaplan–Meier analysis was then performed for T_1_/T_0_ s-CgA and the u-pHPAA cut-off derived from the ROC curves, with Log-rank test results being significant for both T_1_/T_0_ ratios (*p* = 0.001 and *p* < 0.001, respectively) (Appendix A). Cox-regression analysis was performed as well, with no significant results in all the models applied.

ROC curve analyses for disease progression were performed for T_1_/T_0_ s-CgA, T_1_/T_0_ u-pHPAA, T_1_/T_0_ s-NSE, and T_1_/T_0_ u-5-HIAA, showing poor performances (AUC = 0.600, AUC = 0.633, AUC = 0.444, and AUC = 0.600, respectively). Hence, progression-free survival analysis by Kaplan–Meier was not performed.

### 2.3. Metabolomic Profiling: The Role of Tyrosine

The serum metabolic signature and orthogonal partial least squares discrimination analysis (OrthoPLS-DA) (35.1% of the total variance) was performed on the 1H NMR serum spectra with a model accuracy of 75%, suggesting that the T_0_ and T_1_ time point groups are distinctively different in terms of their serum metabolic profiles (Figure 2A).

Interestingly, the score plot shows that the profiles of one patient before (T_0_) and after treatment (T_1_) are different from the other patients. Interestingly, this patient has the highest values of sCGA, NSE, and u-pHPAA. A VIP plot of the top 15 NMR signals reveals that the patients after treatment are characterized by lower plasma levels of ADP, arginine, formate, fumarate, histidine, hypoxanthine, methionine, and pyruvate and higher levels of alanine, asparagine, choline, glutamate, proline, and tyrosine (Figure 2B).

A metabolite-set enrichment analysis based on these metabolites highlighted the complex interplays of several different metabolic pathways and metabolites (Appendix A).

Given that pHPAA is an important metabolite reflecting the impairment in tyrosine metabolism, we decided to specifically evaluate the proton signals of tyrosine at 6.87 ppm. In detail, the tyrosine levels in the serum samples were evaluated at the T_0_ and T_1_ time points, and the T_1_/T_0_ ratio was also reported (Appendix A). The data reported that, at the T_0_ time point, the tyrosine levels were significantly associated with the outcome (*p* = 0.044). Interestingly, the T_0_ tyrosine levels correlated with the T_0_ levels of s-CgA and u-5-HIAA, with *p*-values < 0.0001 and equal to 0.0011, respectively. No significant associations with outcome for the T_1_ and T_1_/T_0_ levels (0.26 and 0.35, respectively) were found.

To determine the optimal cutoff value for serum tyrosine, we performed an ROC curve analysis. We found that tyrosine exhibited a very good performance (AUC = 0.958; Sensitivity: 100%; Specificity: 83%) with a cut-off of 0.204 (Figure 3). Kaplan–Meier analysis was then performed for the T_0_ tyrosine cut-off derived from the ROC curves, with a significant Log-rank test result (*p* = 0.007) (Appendix A).

## 3. Discussion

Despite the fact that a panel of biomarkers is currently utilized in NEN management, the discovery of novel biomarkers is imperative for enhancing diagnosis, guiding therapeutic interventions, and predicting early relapses.

With the aim of addressing this need, we analyzed the serum levels of CgA and NSE and the 5-HIAA and pHPAA levels in 24 h urine samples collected at baseline (T_0_) and after one year of follow-up (T_1_) from fourteen patients affected by NENs.

Interestingly, we confirmed through our preliminary data that urinary para-hydroxyphenylacetic acid (pHPAA), a tyrosine metabolite normally excreted in the urine of healthy individuals, might be a useful marker for managing NET patients.

Our results show that gender was associated with the outcome of death as both non-surviving patients were men, despite the fact that a lack of association was previously reported [16]. Thus, the variable of gender was not included in the statistical analyses due to the limited sample size and the gender disparity.

We found no correlation between the s-NSE levels and metastasis or death, despite its common use in NENs monitoring; nonetheless, its measurement is suggested in NET G2 or G3 in the presence of normal s-CgA levels [12,17,18].

U-5-HIAA is primarily assessed in the diagnosis and follow-up of carcinoid syndrome, according to the ENETS consensus guidelines [7]. In our population, the u-5-HIAA levels were associated with the outcome of death only after 1 year of follow-up, as previously reported in the literature [19].

A possible bias in this study is related to the high biomarker concentrations at baseline in some patients. In order to normalize the data, we calculated the ratios between the 1-year and baseline levels (T_1_/T_0_ parameters), focusing on the increase (or decrease) in biomarker levels, independently from the baseline values, according to the procedures described by Tsy et al. [20].

Although the cohort of patients analyzed was limited, we demonstrated that the baseline levels of S-CgA correlate with metastasis as well as death at T_0_ and T_1_, confirming the previous reports [21,22,23].

Our data confirm the role of s-CgA in the management of NENs patients as well as highlight the possible role of u-pHPAA. Despite the difficulties in the 24 h urine collection, the ROC curve shows the possible clinical utility of u-pHPAA, as proposed in our previous study in 2020 [13]. These findings will nonetheless require confirmation in larger groups of patients, while increasing the frequency of specimen collection as well, in order to calculate the ratios with increased clinical utility. Metabolomics profiling reveals a modulation of the various metabolites at T_1_ compared to the baseline, highlighting the complex interplays of multiple metabolic pathways and metabolites. Interestingly, we identified tyrosine as a significant altered metabolite in these patients, and this result is even more significant if we consider that pHPAA is an important metabolite reflecting an impairment in tyrosine metabolism.

Intestinal bacteria favor the reductive and oxidative metabolism of serum tyrosine and phenylalanine through different pathways that involve 4-hydroxyphenylacetic acid and phenylacetic acid [24], with the former being the direct precursor of *p*-cresol. The production of cresols from tyrosine (and phenylalanine) has been attributed to various intestinal anaerobes, including species of *Clostridioides*, *Bacteroides*, *Bifidobacterium*, and various others [25,26]. On these bases, we hypothesize that the study of the intestinal microbiota in NET patients could further support the role of tyrosine and its metabolites as potential biomarkers in NET tumors. Indeed, the T_0_ serum tyrosine levels showed correlations with both s-CGA and u-5-HIAA and were associated with the outcome of death. We reported a good performance for the T_0_ serum tyrosine ROC curve and significant Log-rank test results following the Kaplan–Meier survival analysis, despite the non-significant results in the Cox regression analysis. Serum tyrosine levels are known to be altered in several type of cancers [27]. A significant decrease in tyrosine levels in plasma samples from gastric cancer patients compared to the controls, as well as between early stage and late-stage gastric cancer, has been reported in the literature [28]. Similarly, patients with esophageal cancer have decreased tyrosine levels in serum compared to healthy controls [29], and a strong association of tyrosine with prostate cancer presence has been reported in the literature as well [30]. Nonetheless, no association is currently reported in the literature between serum tyrosine levels and NENs. A recent paper had explored the metabolomic profile of NENs patients, even though tyrosine was not included in the analyzed profile [15].

The management of NET is still challenging, and extensive efforts are warranted to identify new useful biomarkers for better patient care.

## 4. Materials and Methods

### 4.1. Study Population and Sample Collection

Patients attending the Sarcomas and Rare Tumors Unit of the Istituto Nazionale Tumori-IRCCS-Fondazione “G. Pascale” (Naples, Italy) were enrolled between 2018 and 2019 as part of the IMMUNeOCT trial (EudraCT number 2017-001613-83). Patient recruitment and sample collection were approved by the ethics committee of the National Cancer Institute of Naples-Fondazione G. Pascale. Written informed consent was obtained from all the patients in accordance with the Declaration of Helsinki for the use of human biological samples for research purposes.

The inclusion criteria were age > 18 and <80, diagnosis of NEN < 2 years prior to first specimen collection, and NEN Grading G1 and G2.

Blood and 24 h urine samples were collected at baseline and after 1 year of follow-up, although we should point out that due to the pandemic outbreak in 2020, the timings of the second specimen collection were not always accurate. Patients adhered to the following precise dietary requirements prior to the 24 h urine collection: bananas, vanilla, chocolate, coffee, nuts, vegetables, olives, phenothiazine drugs, and drugs containing gentisic or homogentisic acid (antirheumatic drugs) were restricted. Patients’ clinical conditions were then monitored for at least one more year. Serum samples were analyzed within 3 h from blood withdrawal, and urinary samples were stored at −20 °C while serum samples for metabolomic analysis were stored at −80 °C after centrifugation.

### 4.2. Biochemical Markers

The S-CgA levels were measured on a KRYPTOR compact PLUS (Dasit, Milan, Italy) instrument and the s-NSE levels were measured on a COBAS 6000 (Roche, Basel, Switzerland), according to each manufacturer’s instructions. Concentrations of u-5-HIAA and u-pHPAA were measured in the 24 h urine samples by high-pressure liquid chromatography (Agilent Infinity II, Bio-Rad, Hercules, CA, USA) with electrochemical detection, according to the manufacturer’s instructions. The method for u-pHPAA measurement was previously described [31].

### 4.3. Serum ^1^H-NMR Spectroscopy and Data Processing

All sera samples were prepared for NMR analysis by mixing 300 μL of PBS/H_2_O (10:90) and 70 μL of reference standard D_2_O solution containing 0.1 mM sodium 3-trimethylsilyl [2,2,3,3-2H4] propionate (TSP) with 330 μL of plasma. Samples were inserted into an NMR tube, and all the spectra were recorded using a 600 MHz NMR spectrometer Bruker Avance III HD (599.97 MHz) equipped with a TCI cryoprobe. A standard Carr–Purcell–Meiboom–Gill (CPMG) presaturation pulse sequence was used to decrease the broad NMR signals from the slowly tumbling molecules in the proteins and lipids and to suppress the water peaks. In our experiment, the data points were acquired using 256 transients. All the ^1^H NMR spectra were manually phased and baseline-corrected and then referenced to the CH_3_ resonance of TSP at 0 ppm. The spectral 0.50–8.60 ppm region of the 1H-NMR spectra was integrated by the AMIX package (Bruker, Biospin, Ettlingen, Germany) in buckets of 0.04 ppm, excluding the water resonance region (4.5–5.2 ppm), and normalized to the total spectrum area using Pareto scaling by the MetaboAnalyst v5.0 tool [32]. We used, as a reference, the proton signal of the tyrosine at 6.87 ppm because it was not overlapped with the proton signals of other metabolites.

### 4.4. Statistical Analysis

Descriptive statistics included the median (interquartile range) for variables with data that were not normally distributed. Comparisons between the groups were analyzed with a two-tailed Mann–Whitney U test as all of the continuous variables were not normally distributed. Pearson’s correlation analysis between the proton signals of tyrosine and other biochemical markers was performed. ROC curves were outlined in order to evaluate the performance of different biomarkers, using the outcome of death as the state variable. The ROC curves were classified depending on the AUC value as “poor” (AUC < 0.900), “good” (AUC ≥ 0.900 and <1.00), or “excellent” (AUC = 1.00). Kaplan–Meier survival analysis and Log-rank test were carried out. Cox regression analysis was performed with the following three different models: Model A (T_0_/T_1_ s-CGA and u-pHPAA), Model B (T_0_/T_1_ s-CGA and u-pHPAA; treatment with somatostatin analogues), and Model C (T_0_/T_1_ s-CGA and u-pHPAA; treatment with somatostatin analogues; disease progression during follow-up). Statistical analyses were performed with Statistical Package for Social Science (SPSS Inc., Chicago, IL, USA), version 28.0.

## 5. Conclusions

In conclusion, our study highlights the role of u-pHPAA as a novel biomarker, corroborated by the results obtained from the measurement of serum tyrosine by ^1^H-NMR spectroscopy. Our findings suggest that, once confirmed in larger studies, the measurement of u-PHPAA adds to the commonly used biomarkers such as s-CgA and might improve the management of NENs.

## Figures and Tables

**Figure 1 ijms-25-12317-f001:**
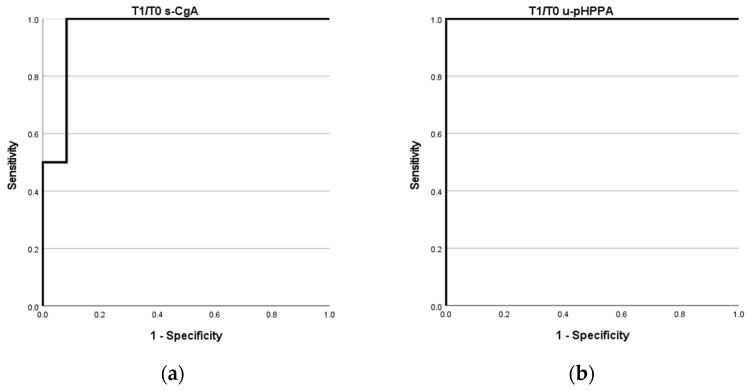
(**a**) T_1_/T_0_ serum chromogranin A (s-CGA) ROC curve; (**b**) T_1_/T_0_ urinary para-hydroxyphenylacetic acid (u-pHPPA) ROC curve.

**Figure 2 ijms-25-12317-f002:**
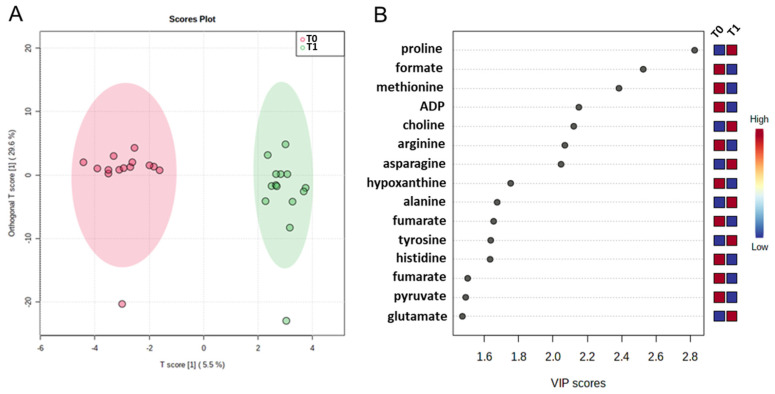
Score plot (**A**) and Variable Importance Plot (VIP) (**B**) related to metabolomic profiling on plasma samples at baseline (T_0_) and after one year of follow-up (T_1_). In the score plot, the red and green circles indicate the metabolome evaluated for each of the fourteen patients at T_0_ and T_1_ time. In the VIP plot, the colored boxes on the right from blue to red indicate low and high levels of the metabolites in each group under study (T_0_ and T_1_), respectively. Higher VIP scores indicate more difference between the metabolite levels at two times.

**Figure 3 ijms-25-12317-f003:**
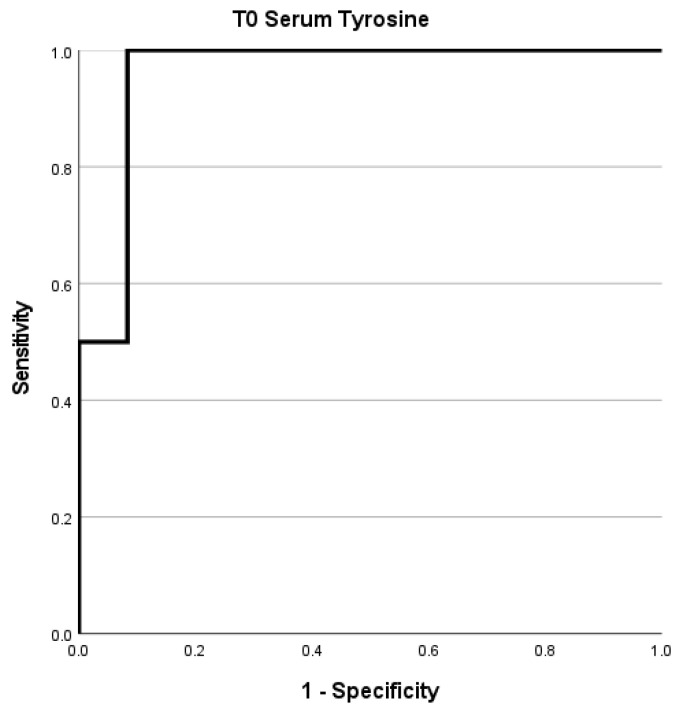
T_0_ serum tyrosine ROC curve.

**Table 1 ijms-25-12317-t001:** Population characteristics.

Characteristics	Patients (n = 14)
Age, median (IQR)	53 (40–61)
Gender, n (%)	
Female	11 (79%)
Male	3 (21%)
Localization, n (%)	
Breast	1 (7.1%)
Extrasurrenalic Paraganglioma	1 (7.1%)
Gastrointestinal	8 (64%)
Lung	1 (14%)
Ovary	1 (7.1%)
BMI, n (%)	
<25	9 (64%)
>25	5 (36%)
Metastasis, n (%)	
No	6 (43%)
Yes	8 (57%)
Outcome, n (%)	
Alive	12 (85.7%)
Dead	2 (14.3%)

**Table 2 ijms-25-12317-t002:** Biochemical markers at baseline (T_0_), after 1 year of follow-up (T_1_), T_1_/T_0_ ratio, and patients’ outcome.

Patient ID	u-5-HIAA (mg/24 h)			u-pHPAA (µmol/mmol Ucreat)			s-CgA (ng/mL)			s-NSE (ng/mL)			Outcome
	T_0_	T_1_	T_1_/T_0_	T_0_	T_1_	T_1_/T_0_	T_0_	T_1_	T_1_/T_0_	T_0_	T_1_	T_1_/T_0_	
1	1.6	3.8	2.37	13.0	10.0	0.77	72.5	66.7	0.92	17.2	15.9	0.92	No
2	1.7	3.7	2.18	5.0	5.0	1.00	133.6	640.4	4.79	10.5	10.0	0.95	No
3	66.0	125.7	1.90	11.0	27.0	2.45	593.5	407.0	0.69	15.1	13.7	0.91	DP
4	1.5	3.1	2.07	21.0	13.0	0.62	32.3	26.0	0.81	18.7	16.8	0.90	DP
5	2.6	11.6	4.46	10.0	36.0	3.60	74.7	25.7	0.34	12.5	15.3	1.22	No
6	109.7	130.0	1.19	19.0	133.0	7.00	4550.0	6862.0	1.51	15.4	17.4	1.13	Death
7	3.9	2.6	0.67	5.0	7.0	1.40	43.0	63.1	1.47	11.9	18.5	1.55	DP
8	2.7	5.2	1.96	5.0	7.0	1.40	380.3	113.0	0.30	32.4	25.6	0.79	DP
9	2.3	3.8	1.65	23.0	5.0	0.22	735.0	238.0	0.32	9.7	10.5	1.08	DP
10	4.2	0.8	0.19	5.0	15.0	3.00	130.9	26.3	0.20	15.4	28.5	1.85	DP
11	3.5	5.6	1.60	6.0	11.0	1.83	124.4	30.7	0.25	11.9	12.5	1.05	No
12	3.7	1.9	0.51	3.0	4.0	1.33	84.4	71.7	0.85	16.4	15.1	0.92	No
13	4.6	2.1	0.46	35.0	145.0	4.14	269.0	58.2	0.22	11.7	17.9	1.53	DP
14	4.4	16.5	3.75	7.0	89.0	12.71	953.3	31890.0	33.45	12.6	56.0	4.44	Death

u-5-HIAA: urinary 5-hydroxindolacetic acid; u-pHPAA: Urinary para-hydroxyphenylacetic acid; s-CgA: serum Chromogranin A; s-NSE: serum Neuro-Specific Enolase; DP: Disease progression.

## Data Availability

The original data presented in the study are openly available in Zenodo at https://zenodo.org/records/13149543, (accessed on 7 November 2024).

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
