# Peer review of "Assessing Urinary Para-Hydroxyphenylacetic Acid as a Biomarker Candidate in Neuroendocrine Neoplasms"

_ijms, 2024, doi:10.3390/ijms252212317_

Round 1

Reviewer 1 Report

Comments and Suggestions for Authors

The manuscript by Renato de Falco et al. raises important question about the diagnostics and management of neuroendocrine neoplasms (NENs). The main aim of the study was to evaluate u-pHPAA and tyrosine as novel biomarkers.

Major concerns:

The study has significant limitations. Due to small sample size and non-homogeneous patient cohort in terms of NEN localization, specific methods of statistical analyses are needed. The authors should consider taking statistician as co-author of the manuscript for more accurate statistical evaluation of entire data and conclusions. As representative examples:

In Table 1 only descriptive statistics should be applied; in this case, group comparison and the corresponding p value is not applicable.

For Kaplan-Meier estimates, the cut off value equal to 1.49 for patient stratification to the low- and high-risk groups needs an explanation; the term “non-parametric distribution” should be rephrased to “data not normally distributed”. In addition to Kaplan-Meier estimates, hazard ratios and corresponding 95% confidence intervals, estimated by Cox regression models, would be valuable for the survival analysis.

All figure legends should include more information for understanding of the figure content.

The authors should include all data into the main body of the manuscript or to the supplement and avoid the “data not shown” sentence.  

Comments on the Quality of English Language

English editing is recommended in terms of correct scientific wordings.

Author Response

Thank you very much for taking the time to review this manuscript. Please find the detailed responses below and the corresponding revisions highlighted in the re-submitted files.

Comment 1: The authors should consider taking statistician as co-author of the manuscript for more accurate statistical evaluation of entire data and conclusions.

Response 1: Dr Susan Costantini, first co-author of this article, has recently achieved an 2nd level master in Statistical Methods for Clinical Research and Epidemiology. Nonetheless, we are truly grateful for your insights, and we provided to extensively modify our paper.

Comment 2: In Table 1 only descriptive statistics should be applied; in this case, group comparison and the corresponding p value is not applicable.

Response 2: Thank you for your comment, we provided by changing the whole structure of Table 1.

Comment 3: For Kaplan-Meier estimates, the cut off value equal to 1.49 for patient stratification to the low- and high-risk groups needs an explanation;

Response 3: Thank you for your comment. The cut-offs (1.49 for T1/T0 s-CGA and 5.57 for T1/T0 u-pHPAA) are the values at ROC Curve coordinates with best Sensitivity and Specificity. We modified the text in order to clarify this point.

Comment 4: The term “non-parametric distribution” should be rephrased to “data not normally distributed”

Response 4: Thank you, we modified the text as suggested.

Comment 5: In addition to Kaplan-Meier estimates, hazard ratios and corresponding 95% confidence intervals, estimated by Cox regression models, would be valuable for the survival analysis.

Response 5: Thank you for your suggestion, we performed Cox regression analysis but we didn’t find significant results in three different models, as detailed in the Methods section. We modified the text in the revised manuscript.

Comment 6: All figure legends should include more information for understanding of the figure content.

Response 6: Thank you for your comment, we provided to write more detailed legend in Figure 2.

Comment 7: The authors should include all data into the main body of the manuscript or to the supplement and avoid the “data not shown” sentence.  

Response 7: Thank you for your suggestion, we provided to include the not shown data (T1/T0 s-Tyrosine) in Supplementary Table 2.

Reviewer 2 Report

Comments and Suggestions for Authors

The authors have tried to identify novel non-invasive biomarkers for neuroendocrine neoplasms. This is a good endeavour with higher translational potential. In spite of the patient data, this can be validated in vitro and in vivo and in addition studied for specific inhibitors that can control the disease in a therapeutic perspective.

Comments on the Quality of English Language

The authors have tried to identify novel non-invasive biomarkers for neuroendocrine neoplasms. This is a good endeavour with higher translational potential. In spite of the patient data, this can be validated in vitro and in vivo and in addition studied for specific inhibitors that can control the disease in a therapeutic perspective.

Author Response

Thank you very much for taking the time to review this manuscript.

Comment 1: In spite of the patient data, this can be validated in vitro and in vivo and in addition studied for specific inhibitors that can control the disease in a therapeutic perspective.

Response 1: Thank you very much for your insights, we aim to perform more in vivo studies on a larger population in order to validate our data.

Reviewer 3 Report

Comments and Suggestions for Authors

The article by de Falco et al. investigates urinary para-hydroxyphenylacetic acid (u-pHPAA) as a biomarker for NENs, aiming to improve patient management by supplementing current biomarkers like CgA, NSE, and 5-HIAA. Through baseline and one-year follow-up measurements, the study finds associations between elevated u-pHPAA and worse outcomes, proposing it alongside serum tyrosine as markers.

Major points: 

The exploration of u-pHPAA as a biomarker could be a useful addition to NEN diagnostics.

The study applies ROC curve analysis, adding robustness to their findings, with survival analyses suggesting the validity of u-pHPAA and tyrosine as potential indicators.

However, with only fourteen patients, the study is severely underpowered, impacting generalizability. Gender imbalance also limits u-pHPAA and tyrosine as reliable markers for broader populations. Two out of 14 patients did not survive after one year.

The patients are from the IMMUNeOCT trial, “Octreotide LAR in the induction of immunologic response in patients with neuroendocrine tumors.” Oversight on the treatment regimen and individual patient response data are necessary. Please include a graphical overview showing each patient's absolute and relative measurements and correlations with outcome, even if negative, for clinical relevance.

The utility of the T1/T0 ratio for diagnosis is unclear, because it assesses longitudinal changes rather than baseline diagnostics. Clinical utility is restricted by the need for a one-year follow-up, which limits early diagnostic value. The study lacks data on correlations between T1/T0 ratio changes and treatment response, essential for assessing whether u-pHPAA reflects disease activity or unrelated factors. Response and additional outcomes, such as PFS, correlations with tumor metrics would support the clinical relevance.

Dietary restrictions, such as avoiding olives during urine collection, should be clarified to control dietary interference.

The title's reference to "management" may be misleading and is unsubstantiated here.

Please make clear in Figure S2: Are the differences in the survival curves significant for the high cutoff level for T1/T0 versus the low one? 

A more accurate title might be “Assessing urinary para-hydroxyphenylacetic acid as a biomarker candidate in neuroendocrine neoplasms.”

Minor points: 

In Table 2, replace commas with periods (e.g., 3.75 for patient #14).

Abstract: Correct to "p=0.044," and revise “good” ROC performance to a consistent rating. Provide a consistent scoring of your ROC curve values.

Use "fourteen" instead of "Fourteen" in the Discussion.

"Intestinal bacteria favors" should be “favor” to align with plural usage.

The sentence “Despite the difficulties of the 24-hour urine collection, mostly related to dietary requirements, the ROC curve shows the clinical utility of u-pHPAA, as proposed in our previous study in 2020…” is misleading. Please clarify, remove “mostly related to dietary requirements,” or say, “Despite variances of the 24-hour…”

Recommendation: Accept with major revisions.

Author Response

Thank you very much for taking the time to review this manuscript. Please find the detailed responses below and the corresponding corrections highlighted in the re-submitted files.

Comment 1: Oversight on the treatment regimen and individual patient response data are necessary.

Response 1: Thank you for your suggestion, we modified the text reporting patients’ treatment regimen in the Results section and in Supplementary Table 1.

Comment 2: Please include a graphical overview showing each patient's absolute and relative measurements and correlations with outcome, even if negative, for clinical relevance.

Response 2: Thank you for your suggestion. We reported each measurement in Table 2 and we added an Outcome column in order to appoint patients’ death or disease progression.

Comment 3: The utility of the T1/T0 ratio for diagnosis is unclear, because it assesses longitudinal changes rather than baseline diagnostics. Clinical utility is restricted by the need for a one-year follow-up, which limits early diagnostic value.

Response 3: Thank you for your comment. We are aware that the T1/T0 ratios have unclear utility in clinical terms, and for this reason we appointed in the revised text that more frequent sample collections are needed to verify this aspect.

Comment 4: The study lacks data on correlations between T1/T0 ratio changes and treatment response, essential for assessing whether u-pHPAA reflects disease activity or unrelated factors.

Response 4: Thank you for your suggestion, we performed ROC Curves for T1/T0 s-CGA, T1/T0 u-pHPPA, T1/T0 s-NSE and T1/T0 u-5HIAA T for progression disease. We reported this information into the revised text.

Comment 5: Response and additional outcomes, such as PFS, correlations with tumor metrics would support the clinical relevance.

Response 5: Thank you for your suggestion, we implemented our work with progression disease analysis. No correlation was found for progression disease with analyzed biochemical markers, thus we wouldn't be able to perform PFS analysis. We reported these results in the revised text.

Comment 6: Dietary restrictions, such as avoiding olives during urine collection, should be clarified to control dietary interference.

Response 6: Thank you for your suggestion, we modified our text in order to detail the dietary restriction patients followed prior to 24-urine specimen collection.

Comment 7: The title's reference to "management" may be misleading and is unsubstantiated here. A more accurate title might be “Assessing urinary para-hydroxyphenylacetic acid as a biomarker candidate in neuroendocrine neoplasms.”

Response 7: Thank you for your insight, we changed the title as proposed.

Comment 8: Please make clear in Figure S2: Are the differences in the survival curves significant for the high cutoff level for T1/T0 versus the low one? 

Response 8: Thank you for your questions, differences in survival for patients with values above and below cutoff value are significant. We added the Log Rank test results to the image as well, in order to clarify this aspect.

Comment 9: Minor points

Response 9: Thank you for your corrections, minor points have been dealt with and corrected in the revised paper. In particular, we detailed the ROC curve scoring system in the Methods sections.

Round 2

Reviewer 3 Report

Comments and Suggestions for Authors

The authors have improved the quality of the paper. 

I appreciate that they followed my suggestions for the title.

However, I recommend splitting 'biomarker' differently, as 'bi-omarker' appears weird.